# Time Series Surface Temperature Prediction Based on Cyclic Evolutionary Network Model for Complex Sea Area

**Jiahao Shi** [1,†]**, Jie Yu** [1,†]**, Jinkun Yang** [2,]*****, Lingyu Xu** [1,3,]***** and Huan Xu** [1]

1 Department of Computer Engineering and Science, Shanghai University, Shanghai 200444, China; shijiahao@shu.edu.cn (J.S.); jieyu@shu.edu.cn (J.Y.); huanxu@shu.edu.cn (H.X.)
2 National Marine Data and Information Service, Tianjin 300171, China
3 Shanghai Institute for Advanced Communication and Data Science, Shanghai University, Shanghai 200444, China
* Correspondence: yangjinkun2021@163.com (J.Y.); xly@shu.edu.cn (L.X.)
† These authors contributed equally to this work.

**Abstract:** The prediction of marine elements has become increasingly important in the field of marine research. However, time series data in a complex environment vary significantly because they are composed of dynamic changes with multiple mechanisms, causes, and laws. For example, sea surface temperature (SST) can be influenced by ocean currents. Conventional models often focus on capturing the impact of historical data but ignore the spatio–temporal relationships in sea areas, and they cannot predict such widely varying data effectively. In this work, we propose a cyclic evolutionary network model (CENS), an error-driven network group, which is composed of multiple network node units. Different regions of data can be automatically matched to a suitable network node unit for prediction so that the model can cluster the data based on their characteristics and, therefore, be more practical. Experiments were performed on the Bohai Sea and the South China Sea. Firstly, we performed an ablation experiment to verify the effectiveness of the framework of the model. Secondly, we tested the model to predict sea surface temperature, and the results verified the accuracy of CENS. Lastly, there was a meaningful finding that the clustering results of the model in the South China Sea matched the actual characteristics of the continental shelf of the South China Sea, and the cluster had spatial continuity.

**Keywords:** time series; deep learning; data mining; prediction

## 1. Introduction

As the ocean is treasured by human beings, all countries worldwide have attached much importance to marine research. The prediction of marine elements plays a significant role in various ocean-related fields such as marine fisheries, marine aquacultures, and maritime communications [1–3]. Among these ocean-related elements, sea surface temperature (SST) plays a vital role in regulating climate and its changes [4]. Therefore, it is well suited to monitoring climate change. Various climate phenomena such as ENSO(El Niño-Southern Oscillation) are closely associated with the changes in sea surface temperature [5]. In addition, SST could have a significant effect on the growth of marine organisms and the distribution of fishery resources [6,7]. In this sense, it is vitally important to predict the sea surface temperature in an accurate and effective way.

Many methods have been proposed to forecast SST. However, the time series data in SST involve complex coupling mechanisms and spatio–temporal relationships, thus making the data vary significantly in some specific time series. Moreover, the rules of data in different spaces also differ, which makes predictions of events challenging. For example, the sea surface temperature is affected by the external environment, such as ocean currents and salinity. The external environment causes uncertainty in the prediction of sea temperature. Conventional models often predict the global characteristics of the data but

ignore the existence of the special data, which leads to some inaccurate predictions. Because the predictive analysis using a single prediction model for the entire dataset often obtains the overall characteristics of all data rather than training the predictions in a more targeted way, the data errors in the single prediction model are different. The data with a smaller error, mean that the characteristics of this part of the data are similar to the characteristics of the overall data. On the contrary, the data with a larger error, mean that the characteristics of this part of the data are inconsistent with the overall characteristics.

Based on this analysis, a mechanism is required to train different models on the data with different characteristics. In this paper, we learn from the idea of ensemble learning to propose a cyclic evolutionary network model (CENS), which trains the original dataset in a data-driven manner. Since the traditional prediction models obtain different errors when predicting ocean elements at different times and locations, this model uses the prediction error of the traditional prediction model to extract the data with different features. In addition, the model trains several networks to form a network group for prediction, thus avoiding the problem that ensemble learning methods cannot handle special data with a single model. The model divides the data into multiple stages through training errors, partitioning a dataset with different features each time, and eventually training all the datasets with their corresponding networks. Different regions of data can be matched to different networks, which can improve the prediction accuracy of the model. At the same time, a cluster can be constructed from the results of multi-model data classification. The contributions of the paper are detailed as follows:

The proposed model can identify the characteristics of data in complex sea areas. By using the prediction error of the network, the model is able to extract implicit features from the data.

The model is a multi-network model obtained by means of data partition. Unlike traditional methods, which require such methods as DTW (Dynamic Time Warping) to segment the data, this is an end-to-end model. The data from different regions can be automatically selected to the corresponding networks. Multiple networks make predictions, respectively, for the data with different regions, which improves the accuracy of predictions.

The effectiveness of the model has been fully verified in actual data. We experimented with data from the Chinese Bohai Sea and the South China Sea; the model results were better than other models in mean absolute error and root mean squared error.

## 2. Related Work

Our work is mainly related with two lines of research: SST prediction and ensemble learning method. On this basis, the related work can be summarized into two parts.

### 2.1. SST Prediction

The methods of SST prediction are classified into two categories [8]. One is the numerical-based method, which relies on multiple physical formulas to make predictions. With a large number of elements other than SST as parameters, it predicts SST through correlation analysis. For example, after discovering a potentially useful predictive relationship between winter North Atlantic Sea Surface Temperature Anomaly (SSTA) and the subsequent summer (July–August) Central England Temperature (CET), Colman used it to build a regression equation for predicting subsequent summer CET based on the strength of the eigenvectors [9]. In general, this method requires the use of external elements in large numbers but is low in the accuracy of prediction, as a result of which it is rarely used in practice to predict SST.

The other is a data-driven method, which does not require any prior knowledge but has the capability to learn the rules from the input data automatically. This kind of method is dominated by machine learning and deep learning. Machine learning methods, such as genetic algorithms, support vector machines, and Markov chains, have produced excellent time series prediction results [10–12]. For example, He et al. [13] proposed the SSTP model, which relies on dynamic time warping (DTW) to mine the similarity of the historical SST

series and then predicts SST via SVM, while Lee et al. [14] applied a multilevel vector autoregressive model (VAR-L) to achieve a satisfactory result in forecasting sea surface temperature. In general, these machine learning models outperform the numerical-based method in accuracy and simplicity.

In recent years, deep learning has experienced rapid development. Due to its capability to extract complex features from data, neural networks are widely applied in various fields. Recurrent neural networks (RNNs), long short-term memory units (LSTMs), and gated recurrent units (GRUs) [15–17] have been successful in time series forecasting because they can capture nonlinear relationships in time series data. Due to the effectiveness of these models in time series forecasting, neural networks are also commonly used in marine science. For example, Zhang et al. [18] applied a gated recurrent unit (GRU) neural network algorithm to predict SST, which led to good results. Xu et al. [19] combined a convolutional neural network (CNN) and LSTM to extract complex features, and the resulting model had high generalization ability. Xie et al. [20] used the attention mechanism to predict the surface temperature of the ocean, and the model had an excellent performance in terms of long-scale and long-term predictions.

Despite the high accuracy reached by the above-mentioned deep learning models, the data of the entire prediction area are inputted into the model when predictions are made, which ignores the special data, thus leading to the inaccuracy of predictions.

*2.2. Ensemble Learning*

In general, the aforementioned methods involve a single model method. In addition, some scholars adopt ensemble learning methods to make predictions. In respect of ensemble learning, there are two main methods applicable for model selection. One is heteromorphic ensemble learning [21,22], which requires the use of different learners for integration. Its representatives are the superposition method [23] and Yuan learning method [24]. The other is homomorphic ensemble learning, which involves the use of the same learner for integration. Homomorphic ensemble learning includes naive Bayes integration, decision tree integration [25], and neural network integration [26–28].

In recent years, due to the extensive application of neural networks, there has been an effort made to integrate neural networks. For example, Livieris et al. [29] adopted three of the most widely used strategies of ensemble learning for forecasting the hourly prices of major cryptocurrency: ensemble-averaging, bagging, and stacking with advanced deep learning models. Yang et al. [30] put forward the deep neural network ensemble method to model and predict the Chinese stock market index and took the bagging approach to combine neural networks for generating ensemble, which reduces the generalization error. Based on clustering and co-evolution, Minku et al. [31] proposed a neural network that could improve and maintain accuracy by dividing different regions. Zhang et al. [32] proposed a sample-integrated genetic evolutionary network that employs genetic evolutionary strategies to construct subsample sets, and uses integrated learning strategies to combine multiple models. Yu et al. [33] proposed a multilevel neural network-integrated learning model that can be applied to evaluate the reliable value of the network model for comprehensive output. Lu et al. [34] proposed a new neural network ensemble method by selecting individual networks through diversity measurement and assigning appropriate weights to individual networks. However, the ensemble learning method is aimed at fusing different networks for output, which makes it unable to deal with special samples. Therefore, we proposed a cyclic evolutionary network. It combines the advantages of single networks and ensemble learning and can handle the data in a complex environment.

## 3. The Cyclic Evolutionary Network (CENS)

*3.1. Model Framework*

CENS uses multiple networks to split the data in terms of the error of the prediction results, where the results with different errors represent different data features. This model trains several networks to form a network group for prediction, thereby improving the

accuracy and precision of prediction. As shown in Figure 1, the model consists of two phases: the training phase and the application phase. The training phase is comprised of the single network unit, dataset partition mechanism, and evolutionary termination mechanism. The single network unit trains the data and obtains the test error. The error of each data element is different due to the various characteristics of the data. The dataset partition mechanism splits the original data into an evolution set and an elimination set automatically based on the prediction error, thus enabling clustering of the data. The evolutionary termination mechanism is used to determine whether the model stops training. CENS can obtain a single network unit after each round of training. We define a CENS with N single network units as N levels of CENS. Finally, the model will generate networks with several levels adaptively using this mechanism. The application phase and the training phase are relatively independent of each other. In the test phase, the data are first put into the prediction network of each level. Then, the prediction network with the minimum prediction error is selected. Finally, the data are used with the selected network to obtain the final prediction result. The details are shown in Algorithm 1.

---

**Algorithm 1. The Training of CENS**

---

Assume that the region consists of k data grid points
1. Data preprocessing: slice the data through sliding window to obtain data sample.
2. Put the data samples into single network unit for training.
3. Use the dataset partition mechanism to divide data samples into evolution sets and elimination sets.
4. The elimination set is treated as a new data sample to repeat steps 2 and 3 until the evolutionary termination mechanism terminates the training of the network.
5. Finally, the CENS consists of N single network units, the size of which is determined by how many times the steps are repeated.
Note: Considering that the first network captures the features of the overall dataset, the first network is used as a pre-trained model in the later networks, which not only allows the model to fit faster, but also improves the overall accuracy of prediction. The hyperparameters of each single network unit are the same. The difference is that the input samples of each single network unit are different. The first single network unit has the most input samples, and the second single network unit has the second most input samples, and so on.

---

The following parts will introduce the single network unit, dataset partition mechanism, evolutionary termination mechanism, and application phase of the model.

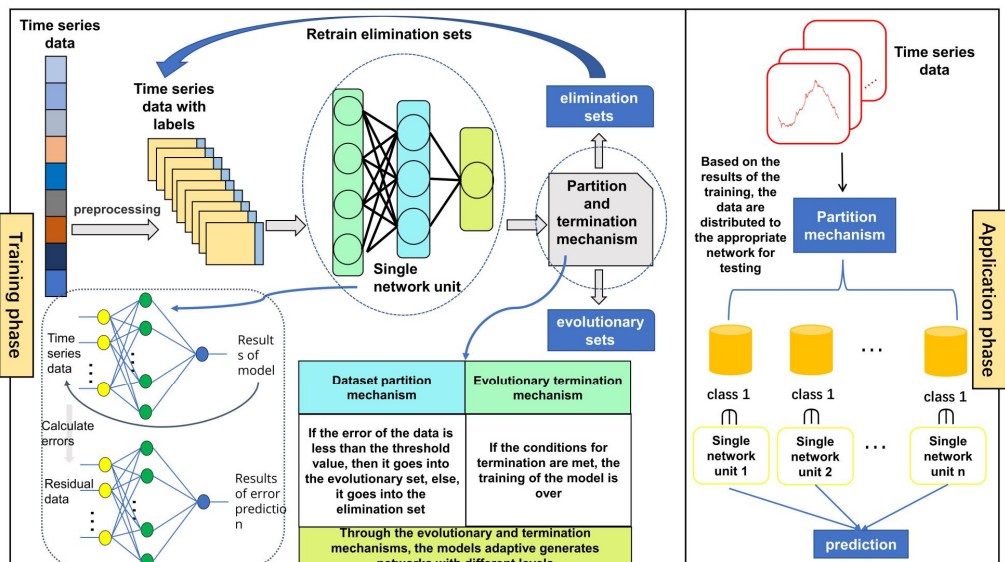

**Figure 1.** Structure of the CENS model.

### 3.2. Single Network Unit

Each unit consists of a numerical prediction network and an error prediction network. The numerical prediction network is used to obtain the absolute error of the data. The error prediction network is used to obtain the difference between the characteristics of the individual data and those of the dataset. The error prediction network is used to predict the error of the sample data in the data prediction network. Combining the data prediction network and the error prediction network, the model can extract the overall characteristics of the data. If the data error is small, the characteristics of the data match the characteristics of the overall data. Conversely, if the data error is large, the characteristics of the data do not match the characteristics of the overall dataset. Therefore, the error can be used as a criterion for data partition in the cyclic evolutionary network.

The prediction data can be obtained through following the data prediction model

$$D'' = f(D') \tag{1}$$

where $f(\cdot)$ represents the data prediction model, $D'$ denotes the original data processed, and $D''$ denotes the prediction data.

We choose the GED (GRU encoder-decoder) model as the data prediction model $f(\cdot)$. The GED is a model of GRU encoder-decoder with SST code and dynamic influence link (DIL) [12]. It has a good effect in predicting sea temperature, but it cannot provide spatial feature information. In comparison, the evolutionary network framework can cluster points with different spatial features. Therefore, the GED model was chosen as the data prediction model to capture the historical temporal features of the SST and the spatial feature information by the evolutionary mechanism of CENS.

The absolute error is calculated by Equation (2)

$$\Delta X = |D'' - D'| \tag{2}$$

After calculating the absolute error, we use these absolute error data to train an error prediction network

$$\Delta X' = e(\Delta X) \tag{3}$$

where $\Delta X'$ denotes the error prediction data. $e(\cdot)$ is the error prediction model, and we also use the GED as the error prediction model.

If the error prediction of the error prediction network is small, the numerical prediction network is considered as able to predict accurately. If the error prediction is large, the numerical prediction network is considered as able to accurately predict. Therefore, the errors of the data provide the basis for the dataset partition mechanism. The structure of the single network is shown in Figure 2.

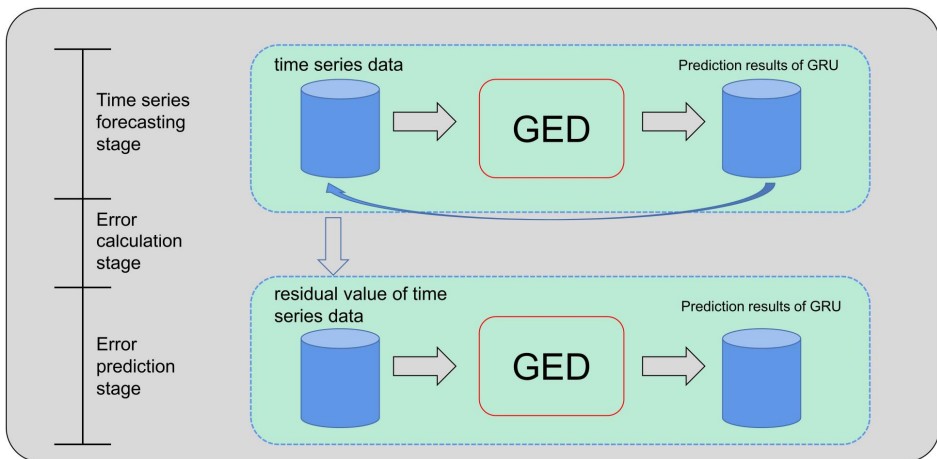

**Figure 2.** Structure of a single network unit. First, the time series data are trained by a GED network [20], and the value predicted by the network is subtracted from the original time series data to obtain the residual data. Then, the residual data are trained by another GED network to obtain the final residual prediction result.

### 3.3. Dataset Partition Mechanism

The dataset partition mechanism is applied to split the dataset into evolution and elimination sets at each level and then move the elimination set into the next level of the network for continued partitioning. The elimination set consists of those data elements whose characteristics do not match those of the majority of the data, as indicated by large prediction errors in that network. Thus, these data need to be processed using other networks. In this way, the dataset partition mechanism divides the raw data into evolutionary and elimination sets based on the error values obtained in the single network units.

After training on a single network, the dataset is divided according to the data partition criteria

$$\text{Data} = g(t_e, \Delta X') \tag{4}$$

where $t_e$ denotes the dividing standard line, $g(\cdot)$ indicates the data partitioning rule that is based on the error mean value. $\Delta X'$ refers to the set of error datasets, and Data denotes the evolution set and elimination set of the original data. Data can be divided into two parts, which can be expressed as

$$\text{Data} = \{E, W\} \tag{5}$$

where $E \cup W = \text{Data}$, $E$ is the evolution set and $W$ is the elimination set. The data partitioning rule can be expressed as Equations (6) and (7)

$$t_e = average(pred\_error) \tag{6}$$

$$g(t_e, \Delta X') = \begin{cases} x \in E & \Delta x < t_e \\ x \in W & \Delta x > t_e \end{cases} \tag{7}$$

Considering that the GED model has a small variance in predicting each point in practical application, and the mean value is more reliable and stable, we choose the average value as the data partitioning rule.

Then, the original dataset can be automatically divided into the evolution set and the elimination set, thus distinguishing the different characteristics of data and dividing the dataset accordingly. The dataset will gradually decrease with the progress of partitioning, and some special data will be left in the end. Therefore, the model will generate the networks with a progressive relationship.

### 3.4. Evolutionary Termination Mechanism

During the phase of cyclic evolutionary network training, the evolution termination mechanism ends the training of the network. In this paper, we propose two termination mechanisms to stop training the model: dataset threshold termination and evolutionary scale termination.

In dataset threshold termination, when the error of the current network on the current dataset is less than the threshold, the network will consider the requirements met and stop training. The termination condition is

$$MAE\left(f^i(T), T\right) < t \tag{8}$$

where $t$ is the error threshold, which can be set based on expert experience. In practical experiments, the $MAE$ value of CENS can be improved by 5–10% on the GED. Thus, in general, we set $t$ to 90% of the $MAE$ value of the GED. $T$ is the dataset.

In evolutionary scale termination, training will be stopped when the proportion of data in the evolution set is greater than rate $R$, which can be set based on training experience. The termination condition is

$$\frac{\sum_{i=0}^{I} num(E_i)}{num(T)} > R \tag{9}$$

where *I* is the level of the current network, *T* is the size of the original dataset, $num(T)$ is the amount of data in the statistical dataset, and $E_i$ is the evolutionary set of *i*.

*3.5. Application Phase of the Model*

The application phase includes two parts: the network selection and prediction process. In the network selection process, the data are used to select the numerical prediction network corresponding to the value with the lowest prediction error based on the output of each level of the error prediction network, which can represent the ability of that level of the network to predict that data element accurately. In the prediction process, the data uses the selected network to obtain the final predictive result. Compared with traditional deep learning methods, this model realizes the transition from model science to data science.

## 4. Experiments

*4.1. Dataset and Hyperparameters*

To test the performance of different methods, we used the data that were provided by the Physical Sciences Division, NOAA/Ocean and Atmospheric Research/Earth System Research Laboratory, Boulder, Colorado, USA. The dataset covered the global ocean from 89°52′30″ S to 89°52′30″ N, 0°7′30″ E to 359°52′30″ E. In addition, the spatial resolution was 0.25° × 0.25°.

The CENS model utilizes the data of the Bohai Sea and the South China Sea. The temperature fluctuations in the Bohai Sea are significant because it is affected by the continental shelf and northern continental climate [35]. Therefore, we choose the data of the Bohai Sea to measure the accuracy of the proposed model. The data contain daily data (4749 days) from 1 January 2004 to 31 December 2016, ranging from 37.125° to 40.875° N and 119.375° to 121.875° E, basically covering the Bohai Sea, China. We selected 16 points, 100 points, and 224 points for the experiment. In addition, we also used 224 points for comparison experiments. The selected area is shown in Figure 3a–c. The experimental ranges of 16 points, 100 points, and 224 points were 37.125°–38.375° N and 119.125°–121.875° E, 37.875°–39.375° N and 119.375°–121.875° E, and 37.125°–40.875° N and 119.375°–121.875° E respectively. Moreover, the South China Sea is an open sea, and ocean currents and other factors influence it [36,37]. We chose the data of the South China Sea. The selected sea area from 1 January 2004 to 31 December 2016. We selected 16 points, 100 points, and 674 points for the experiment. In addition, we also used 674 points for comparison experiments. The selected area is shown in Figure 3d–e. The experimental ranges of 16 points, 100 points, and 674 points were 6.325°–7.375° N and 106.325°–114.875° E, 6.375°–15.375° N and 106.325°–130.375° E, and 6.375°–30.625° N and 106.325°–130.375° E, respectively.

Since CENS is used to process time series data, we need to use a sliding window to slice the data before training.

The raw time series data are as follows

$$\text{T} = \{t_1, t_2, \ldots, t_m\} \tag{10}$$

where $t_i$ is the original data sample, m is the number of samples, and T is the raw time series data.

Set the window length as K and slicing the original data by sliding window can obtain the processed data

$$\text{D} = \begin{bmatrix} t_1 & t_2\ldots & t_k \\ t_2 & t_3\ldots & t_{k+1} \\ \ldots & \ldots & \ldots \\ t_{m+1-k} & t_{m+2-k} & \cdots & t_m \end{bmatrix} \tag{11}$$

where each row is a data sample. We can obtain (m − k + 1) data samples.

As the model can be divided into a single network unit training phase, dataset partition phase and evolutionary termination phase, there are three hyperparameters in CENS. When selecting the hyperparameters, firstly select the hyperparameters randomly and

find the part with better experimental results. Then, use grid search to obtain the optimal hyperparameters. In the single network unit training phase, the window length was set to 30 for a one-day forecast, 45 for a three-day forecast, and 75 for a seven-day forecast, the learning rate was set to 0.002, batch size was set to 128, the cell size of the GRU in the GED model was set to 32, and we chose Adam optimizer for training. In the dataset partition phase, the threshold was set to the average of the error. In the evolutionary termination phase, considering the limited performance improvement of CENS in small area, we set the rate R to 0.9.

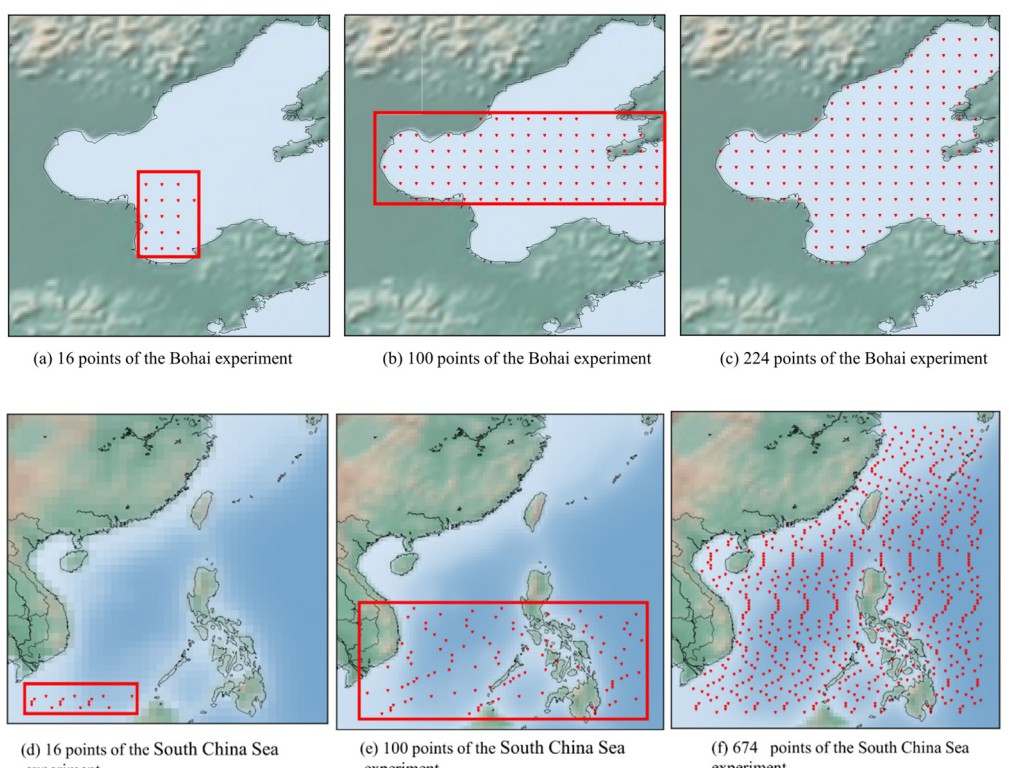

(a) 16 points of the Bohai experiment    (b) 100 points of the Bohai experiment    (c) 224 points of the Bohai experiment

(d) 16 points of the South China Sea experiment    (e) 100 points of the South China Sea experiment    (f) 674 points of the South China Sea experiment

**Figure 3.** Location maps of the experiment. The red dots indicate the predicted points. (**a–c**) represent the predicted areas at 16, 100 and 224 points in the Bohai Sea. (**d–f**) represent the predicted areas at 16, 100 and 674 points in the South China Sea.

### 4.2. Evaluation Metrics

Root mean square error (*RMSE*) and mean absolute error (*MAE*) are mainly used as the criteria for evaluating the quality of the model. *RMSE* is the square root of the square of the difference between the measured value and the true value and *MAE* is the average value of the absolute error. The Equations are as follows

$$RMSE = \sqrt{\frac{1}{n} \sum_{i=1}^{n} \left( X_i^p - X_i^r \right)^2} \tag{12}$$

$$MAE = \frac{1}{n} \sum_{i=1}^{n} \left| X_i^p - X_i^r \right| \tag{13}$$

where $n$ is the number of data, $X^p$ is the prediction of the model. $X^r$ is the true value of the data.

### 4.3. Ablation Experiment

To verify the effectiveness of the framework of the model and the effect of the different levels on the prediction, we conducted experiments on the temperature of the Bohai Sea and the South China Sea. Different levels of the CENS were compared to the experiment. Using the evolutionary termination mechanism, CENS generates 3-levels, 4-levels, and

4-levels networks for predicting 16, 100, and 224 points in the Bohai Sea, respectively. CENS generates 3-levels, 4-levels, and 5-levels networks for 16, 100, and 674 points in the South China Sea, respectively. The results were shown in Figures 4 and 5.

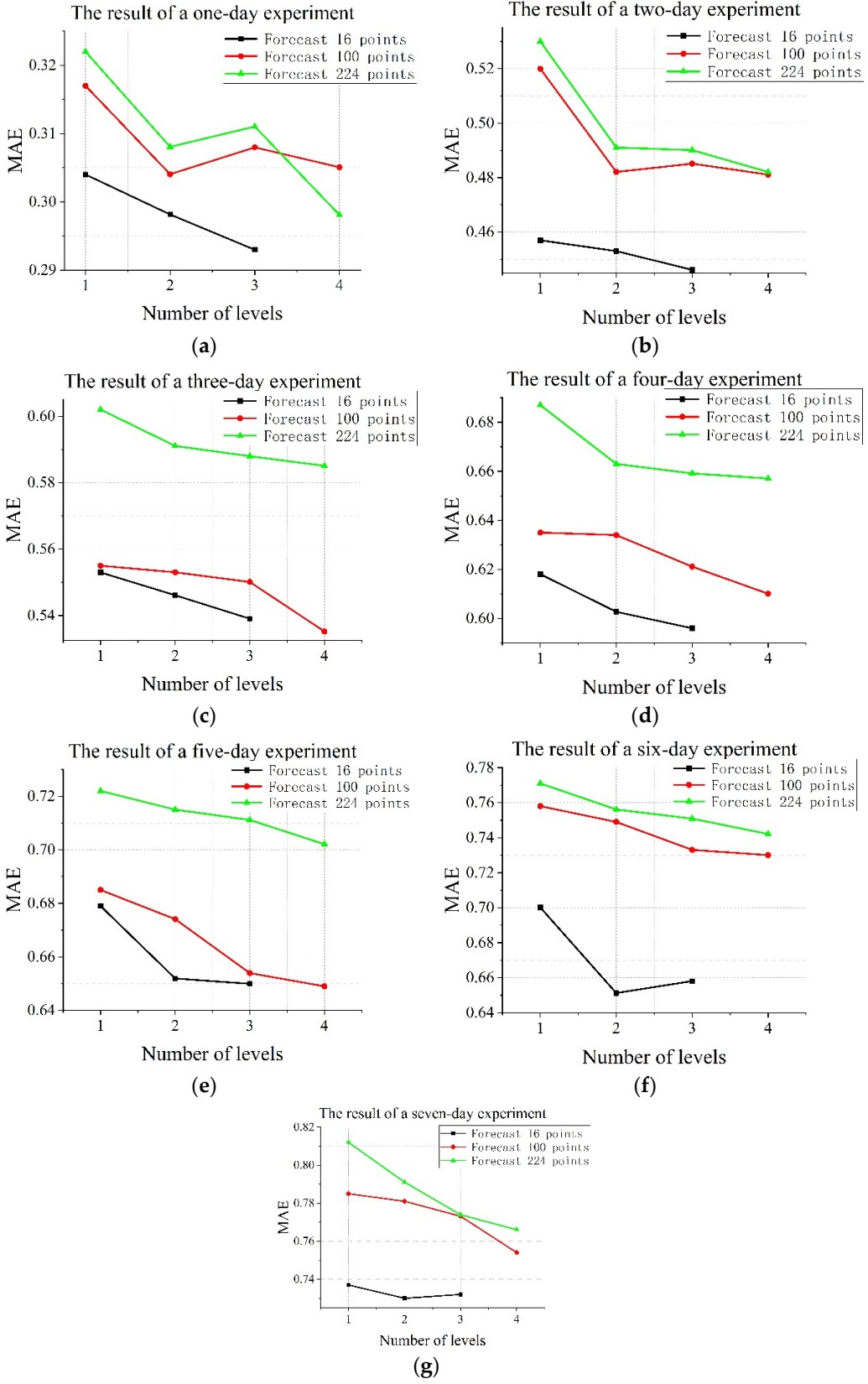

**Figure 4.** Results of CENS in the Bohai Sea for different levels.

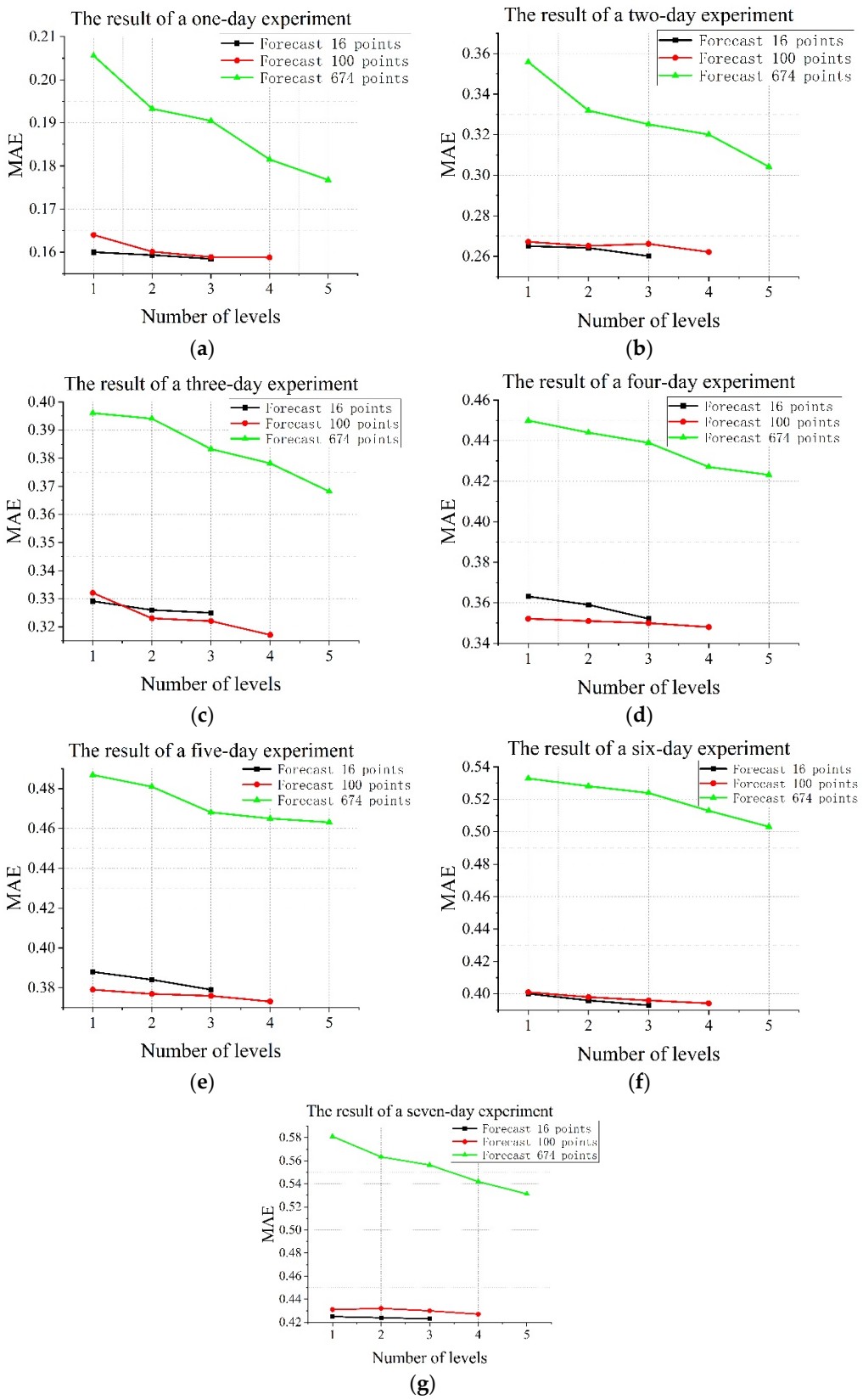

**Figure 5.** Results of CENS in the South China Sea for different levels.

Comparing the experimental results of 16 prediction points in the Bohai Sea and the South China Sea, we can see that when the prediction area is small, the *MAE* and *RMSE* of CENS will generally decrease with the increase in the number of levels, but the improvement is not large.



It means that CENS can capture the spatial dimension information, but the improvement in the prediction of small areas is limited. Comparing the experimental results of 224 points in the Bohai Sea region and 674 points in the South China Sea region, we can see that the *MAE* and *RMSE* of CENS decrease significantly with the increasing number of levels in a larger prediction area, which means that CENS is able to capture the spatial information of a complex sea area well. On the whole, CENS predictions of 1–7 days becomes more accurate as the number of levels increases, which shows that CENS can cope with different sea areas and forecast times and indirectly indicates the effectiveness and robustness of the CENS framework.

### 4.4. BaseLines and Time Series Prediction

In this section, we compare our model with the baselines on the two datasets. The four baselines are as follow:

SVR: Support Vector Regression, which is widely used in time series prediction.

GBDT: Gradient Boosting Decision Tree. This algorithm consists of multiple decision trees, and the results of all trees are accumulated to make predictions. It is the classical algorithm for ensemble learning.

GED [20]: This is a model of GRU encoder-decoder with SST code and dynamic influence link (DIL).

RC-LSTM [19]: This combines the CNN model and LSTM model to study the temporal and spatial correlation of the data, which shows the state-of-the-art performance.

#### 4.4.1. Results and Analysis

To validate the prediction performance of CENS, we experimented on the Bohai Sea SST dataset (224 points) with different prediction scales of 1, 3, and 7 days, respectively. The comparison methods include SVR, GBDT, RC-LSTM, and GED. The results of the experiment are shown in Table 1. The bold items in the table indicate the optimal predictive performance, i.e., the minimum *RMSE* and *MAE*. It was clear that the performance of CENS was better than the other models in all prediction scales, and SVR always performed the worst. *MAE* of CENS was approximately 8% lower than GED in the predicted 1-day experiment, decreasing to 3% at three days of prediction and rising to 6% in the predicted 7-day experiment. In addition, the *RMSE* of CENS was about 7% lower than GED in the 7-day prediction experiment. The experimental results show that, compared with other models, the prediction performance of CENS is improved in different prediction scales at 1, 3, and 7 days.

**Table 1.** Time series prediction results in the Bohai Sea.

| Models | Metrics | Prediction Length (Day(s)) | | |
|---|---|---|---|---|
| | | 1 | 3 | 7 |
| SVR | *MAE* | 0.565 | 0.655 | 0.868 |
| | *RMSE* | 0.512 | 0.846 | 1.345 |
| GBDT | *MAE* | 0.359 | 0.638 | 0.835 |
| | *RMSE* | 0.471 | 0.821 | 1.125 |
| RC-LSTM | *MAE* | 0.340 | 0.598 | 0.810 |
| | *RMSE* | 0.448 | 0.746 | 1.170 |
| GED | *MAE* | 0.322 | 0.602 | 0.812 |
| | *RMSE* | 0.466 | 0.801 | 1.070 |
| CENS | *MAE* | **0.298** | **0.585** | **0.766** |
| | *RMSE* | **0.415** | **0.735** | **1.002** |

To verify the robustness of the model, we also experimented in the South China Sea (674 points) and the results are shown in Table 2. It can be seen that, compared with other models, the performance of CENS is the best in 1, 3, and 7-day sea surface temperature prediction. *MAE* of CENS was 10% lower than GED in the predicted 1-day experiment, 6% at three days of prediction and 6% in the predicted 7-day experiment. In addition, the *RMSE* of CENS remained optimal in the prediction experiments for 1, 3, and 7 days. This shows that CENS can deal with different sea areas.

**Table 2.** Time series prediction results in the South China Sea.

| Models | Metrics | Prediction Length (Day(s)) | | |
|---|---|---|---|---|
| | | 1 | 3 | 7 |
| SVR | *MAE* | 0.400 | 0.507 | 0.712 |
| | *RMSE* | 0.526 | 0.687 | 0.895 |
| GBDT | *MAE* | 0.214 | 0.427 | 0.632 |
| | *RMSE* | 0.286 | 0.566 | 0.834 |
| RC-LSTM | *MAE* | 0.206 | 0.390 | 0.563 |
| | *RMSE* | 0.296 | 0.546 | 0.735 |
| GED | *MAE* | 0.195 | 0.396 | 0.581 |
| | *RMSE* | 0.284 | 0.536 | 0.762 |
| CENS | *MAE* | **0.176** | **0.368** | **0.531** |
| | *RMSE* | **0.241** | **0.478** | **0.687** |

4.4.2. The Error Distribution of SST

This section visualizes the experimental results of the forecast 1, 3, and 7 days to obtain the error distribution of CENS and other models. The results are shown in Figures 6 and 7. Figure 6 shows the error distribution in the Bohai Sea, and Figure 7 shows the error distribution in the South China Sea. The graph circled in red is where CENS predicts better than GED. It can be seen that, compared with other models, CENS is better than other models at all prediction scales and is more accurate than GED in the offshore area in the Bohai Sea experiment. CENS is more accurate than GED in the entire sea area in the South China Sea experiment. This result shows that CENS can predict complex sea areas.

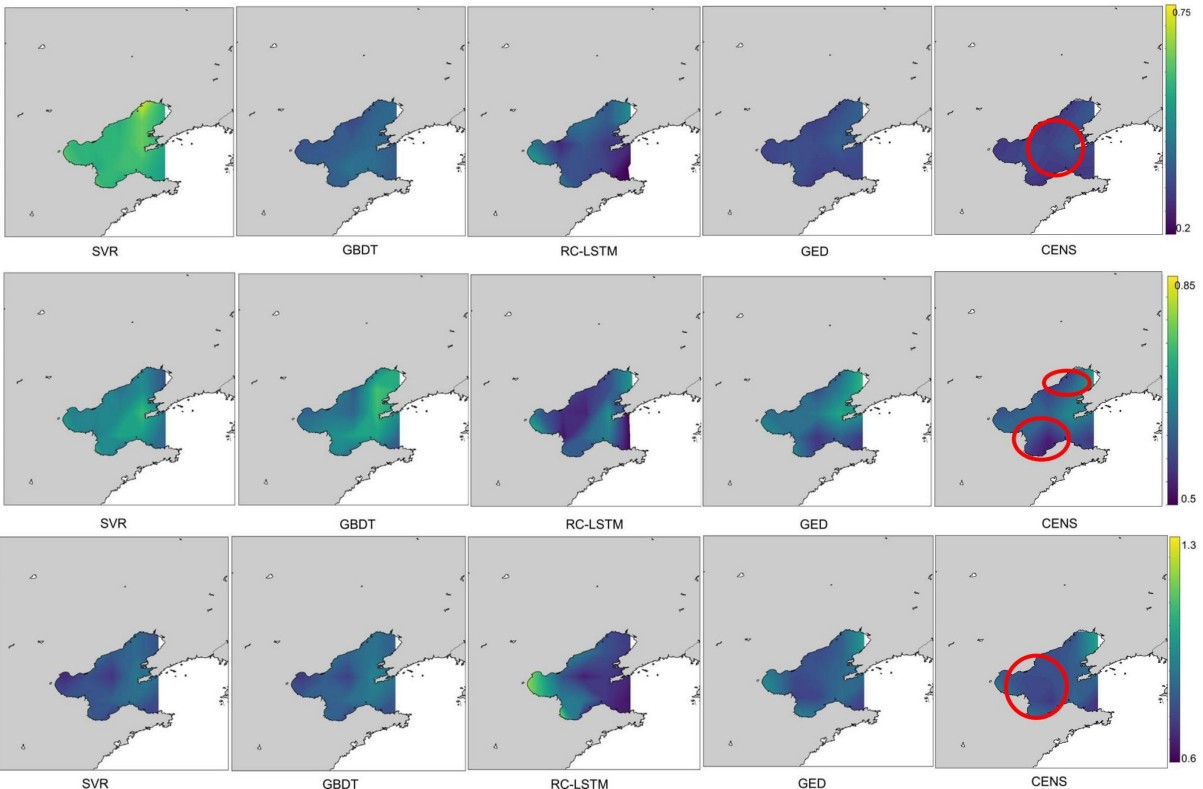

**Figure 6.** Error distribution in the Bohai Sea for different models in 1, 3, and 7-day sea surface temperature prediction. The error plots of SVR, GBDT, RC-LSTM, GED, and CENS are shown from left to right, and the error results of prediction one day, prediction three days, and prediction seven days are shown from top to bottom. The color bar on the right is the *MAE* value.

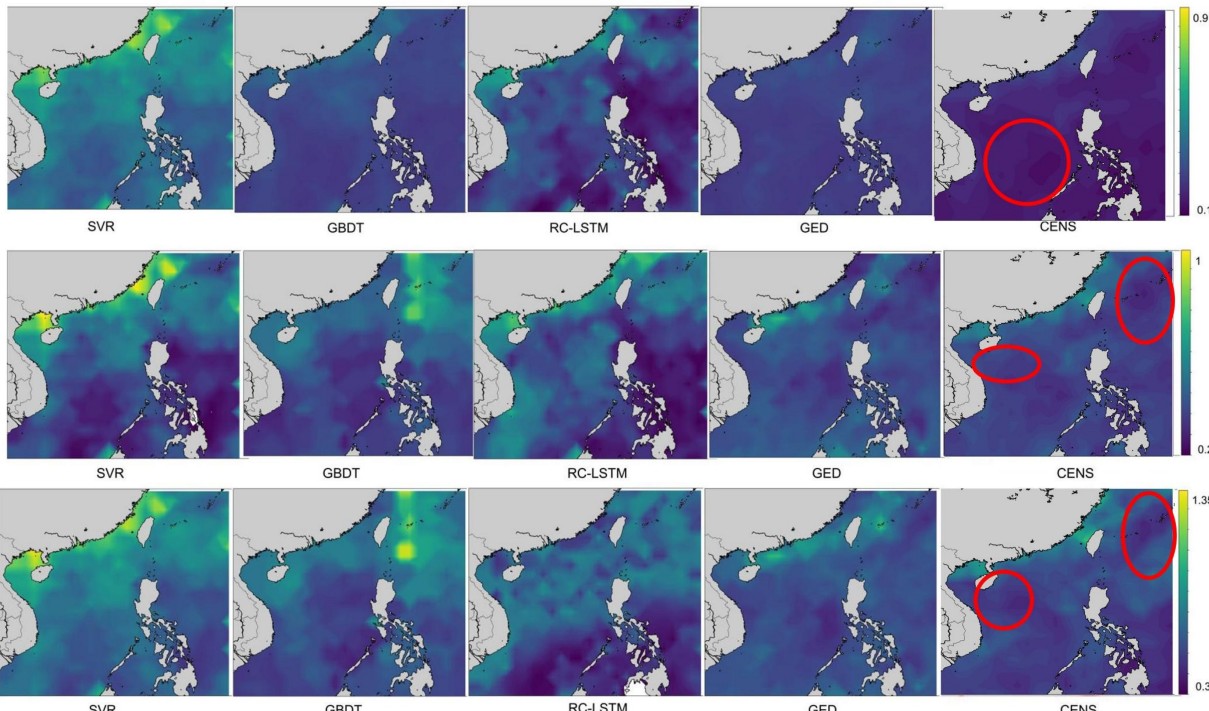

**Figure 7.** Error distribution in the South China Sea for different models in 1, 3, and 7-day sea surface temperature prediction. The error plots of SVR, GBDT, RC-LSTM, GED, and CENS are shown from left to right, and the error results of prediction one day, prediction three days, and prediction seven days are shown from top to bottom. The color bar on the right is the *MAE* value.

### 4.4.3. Computation Efficiency

This section will calculate the elapsed time of each method to test the efficiency of CENS. In the Bohai experiment, we selected 224 points as an input, and the data contained daily data (4749 days) from 1 January 2004 to 31 December 2016. In the South China Sea experiment, we selected 674 points as an input, and the data contained daily data from 1 January 2004 to 31 December 2016. The results are shown in Table 3. SVR is the fastest during the training phase, and CENS is the slowest because CENS is made up of N GED models, and CENS needs to train N GED models. CENS is also the slowest during the test phase because CENS is required to calculate the error network first to obtain the relative data prediction network. Although the speed of CENS is slow, CENS can identify the characteristics of data in complex sea areas, clustering the data automatically, and CENS has a high prediction accuracy.

**Table 3.** The elapsed time of each method.

| Models | Times (s) | Region | |
| :---: | :---: | :---: | :---: |
| | | **Bohai Sea** | **South China Sea** |
| SVR | train | 21.23 | 56.87 |
| | test | 4.72 | 13.13 |
| GBDT | train | 224.77 | 730.74 |
| | test | 0.04 | 0.08 |
| RC-LSTM | train | 767.53 | 843.22 |
| | test | 0.76 | 0.82 |
| GED | train | 905.56 | 964.56 |
| | test | 0.96 | 1.45 |
| CENS | train | 3867.54 | 5732.81 |
| | test | 12.41 | 20.25 |

### 4.5. Discussion

The experimental results show the effectiveness of our proposed model. In this section, we discuss the advantages of the model and the limitations that exist.

#### 4.5.1. The Advantage of CENS

The model shows high accuracy. In practice, the accuracy of the CENS model is higher compared to other baselines, which suggests the capability of the model to extract implicit features from the data.

The model can be relied on to divide and cluster datasets with different characteristics automatically. When used to predict large-scale areas, traditional models routinely rely on DTW and other means to divide the dataset at first, such as RC-LSTM. Differently, our model is an end-to-end model without the need for additional operations, and the classification of the dataset is performed independently. The clustering results of the predicted South China Sea have been visualized. As indicated by the clustering results in Figure 8a and the status of continental shelf visualization in Figure 8b, the clustering boundary region of CENS is highly consistent with the boundary region of the continental shelf shown in Figure 8b in regions 1, 2, and 3. Such areas clustered by evolutionary networks are worthy of research in such ways of correlation analysis.

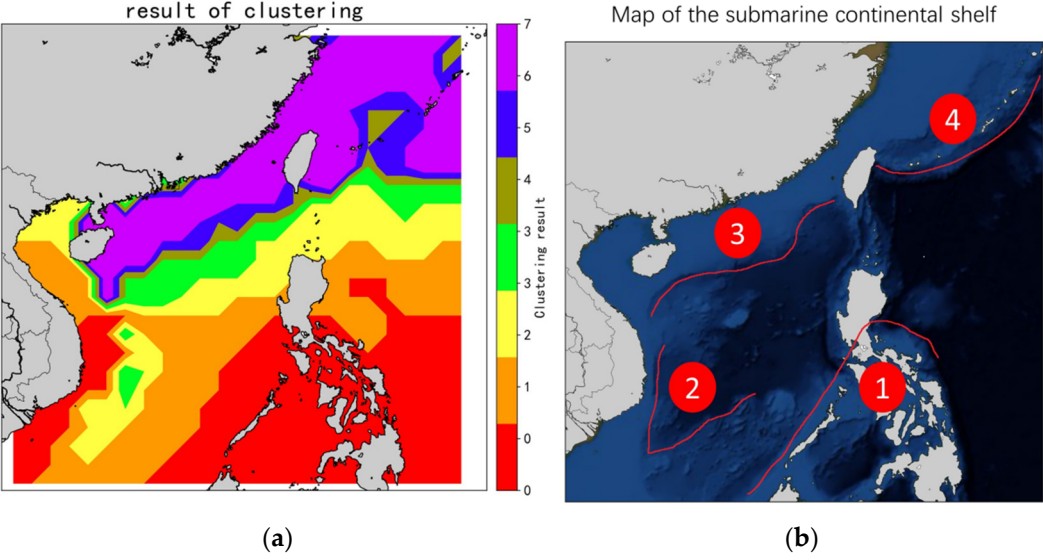

(**a**)　　　　　　　　　　　　　　　　　　(**b**)

**Figure 8.** Cluster result map of CENS and visualization of the submarine continental shelf. (**a**) is the clustering result of CENS. The color bar on the right side of (**a**) is the result of clustering. For example, red is the area of cluster 1, and orange is the area of cluster 2. (**b**) is the submarine continental shelf in the South China Sea. The depth of the color in the (**b**) indicates the water depth.

#### 4.5.2. The Limitations of CENS

Parameter setting. CENS involves many hyperparameters, each of which can affect the prediction results. Therefore, the model necessitates experiments on parameter adjustment in practice.

CENS is unsuited to small datasets. From the results in Figures 4 and 5, it can be seen that there is little room for improving the performance of CENS in predicting small-scale data, and more time is required. This is probably attributed to the fact that the features within the small-scale dataset are more consistent to the extent that they cannot be processed by CENS.

The efficiency of the model is low. Compared with other deep learning models, the CENS model requires several cycles of training. As a result, the training time for CENS is several times longer compared to other models.

## 5. Conclusions

In this paper, a cyclic evolutionary network model (CENS) is proposed. As an error-driven network group, the model is comprised of multiple network node units, with the aim to facilitate training by splitting the dataset with different features. In order to achieve this goal, the prediction error of the model was treated as the division criterion of the dataset, and the dataset was divided automatically. Since different regions of data can be automatically matched to a suitable network node unit for prediction, the model can cluster the data based on their characteristics, thus making it more practical.

Our method is designed to improve the accuracy of the model by automatically clustering the regions with different characteristics in the sea area. Through a comparison with different methods such as SVR, GBDT, GED, and RC-LSTM, it has been verified that CENS can achieve a higher accuracy in SST prediction. In general, this model is a relatively novel and effective solution.

**Author Contributions:** Conceptualization, J.S. and H.X.; methodology, H.X.; software, J.S.; validation, J.Y. (Jie Yu), J.Y. (Jinkun Yang) and L.X.; formal analysis, J.Y. (Jie Yu); investigation, J.Y. (Jinkun Yang) and L.X.; resources, J.Y. (Jinkun Yang) and L.X.; data curation, J.S.; writing—original draft preparation, J.S.; writing—review and editing, J.Y. (Jie Yu); visualization, J.S.; supervision, J.Y. (Jinkun Yang) and L.X.; project administration, J.Y. (Jinkun Yang) and L.X.; funding acquisition, J.Y. (Jinkun Yang) and L.X. All authors have read and agreed to the published version of the manuscript.

**Funding:** This research was funded by the National Program on Key Research Project of China, grant number 2016YFC1401900.

**Data Availability Statement:** Publicly available datasets were analyzed in this study. This data can be found here: https://psl.noaa.gov/.

**Conflicts of Interest:** The authors declare no conflict of interest.

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
