# Peer review of "Time Series Surface Temperature Prediction Based on Cyclic Evolutionary Network Model for Complex Sea Area"

_futureinternet, doi:10.3390/fi14030096_

Round 1

Reviewer 1 Report

1) The data  partitioning rule is based on a threshold ??=???????(????_?????).   (formula 7)

The author should clarify why the average is used rather then a robust measure such as the median. The average may be severely affected by extreme prediction errors.

2) The termination condition is:  ???(??(?),?)<?   Formula 9
"where t is the error threshold, which can be set based on expert experience". In the empirical study, this threshold is set to 0.9.

The author should clarify this point giving some insight on the  rules to follow for choosing this threshold (in general) Moreover, the author should discuss the specific considerations that suggested the selection of the value 0.9 in the empirical applications.

Author Response

Thank you for your  comments.Those comments are very helpful for revising and improving our paper, as well as the important guiding significance to other research. We have studied the comments carefully and made corrections which we hope meet with approval. Please refer to the attachment for specific responses.

Reviewer 2 Report

1. The introduction is not specific to explain how the prediction is used in practice, making the research motivation weak.

2. In Page 5, Line 164
Some symbol is missing in the sentence (noted as ??)
"where f(∙) is the data prediction model, ?′ denotes the original data processed, and ?? denotes the prediction data."
Similar problems should be checked for the whole manuscript.

3. The lines in Figures 3-4 are hard to distinguish. The authors should re-plot these figures with different markers.

4. Some results are not complete for all different levels in Figures 3-4, why?

Author Response

(The authors gave the same response as above.)

Round 2

Reviewer 2 Report

Thanks for the revisions. All concerns are resolved.